

# The AOTF-based $NO_2$ camera

Emmanuel Dekemper[1], Jurgen Vanhamel[1], Bert Van Opstal[1], and Didier Fussen[1]

[1]Royal Belgian Institute for Space Aeronomy, avenue Circulaire 3, 1180 Brussels, Belgium

*Correspondence to:* E. Dekemper (emmanuel.dekemper@aeronomie.be)

**Abstract.** The abundance of $NO_2$ in the boundary layer relates to air quality and pollution sources monitoring. Observing the spatio-temporal distribution of $NO_2$ above well-delimited (flue gas stacks, volcanoes, ships) or more extended sources (cities) allows for several applications: monitoring emission fluxes or studying the plume dynamic chemistry and its transport. So far, most attempts to map the $NO_2$ field from the ground have been made with visible-light scanning spectrometers. Benefiting

from a high retrieval accuracy, they only achieve a relatively low temporal resolution that hampers the detection of dynamic features.

We present a new type of passive remote sensing instrument aiming at the measurement of the 2-D distributions of $NO_2$ slant column densities (SCD) with a high spatio-temporal resolution. The measurement principle has strong similarities with the popular filter-based $SO_2$ camera as it relies on spectral images taken at wavelengths where the molecule absorption cross-

section is different. Contrary to the $SO_2$ camera, the spectral selection is performed by an acousto-optical tunable filter (AOTF) capable of resolving the target molecule's spectral features.

The $NO_2$ camera capabilities are demonstrated by imaging the $NO_2$ abundance in the plume of a coal-fired power plant. During this experiment, the 2-D distribution of the $NO_2$ SCD was retrieved with a temporal resolution of 3 minutes and a spatial sampling of 50 cm (over a $250 \times 250$ m$^2$ area). The detection limit was close to $5 \times 10^{16}$ molecules cm$^{-2}$, with a

maximum detected SCD of $4 \times 10^{17}$ molecules cm$^{-2}$. Illustrating the added-value of the $NO_2$ camera measurements, the data reveal the dynamics of the NO to $NO_2$ conversion in the early plume with an unprecedent resolution: from its release in the air, and for 100 m upwards, the observed $NO_2$ plume concentration increased at a rate of 0.75-1.25 g s$^{-1}$. In joint campaigns with $SO_2$ cameras, the $NO_2$ camera could also help in removing the bias introduced by the $NO_2$ interference in the $SO_2$ measurements.

## 1 Introduction


Nitrogen oxides ($NO_x = NO + NO_2$) play a key role in the air quality of the boundary layer. While NO is produced in combustion processes (transport, thermal power plants, etc.), $NO_2$ mainly appears through the reaction of NO with $O_3$ or $HO_2$. Eventually, the photolysis of $NO_2$ releases an oxygen atom and a NO molecule. To a good approximation, the balance of NO and $NO_2$ is kept constant through this cycle of photo-chemical reactions, which substantiates the widespread use of the $NO_x$

family concept (Seinfeld and Pandis, 2006). Considering the relative ease of measuring $NO_2$ with visible-light spectroscopy, $NO_x$ budgets are often inferred based on $NO_2$ measurements and the photochemical equilibrium assumption.



The most common $NO_2$ remote sensing techniques rely on the differential optical absorption spectroscopy (DOAS), which is based on the fitting of radiance spectra with the effective absorption cross-section of interfering species (e.g. Lohberger et al., 2004). If equipped with a 2-D sensor array, these instruments disperse the light spectrum along one dimension and record its spatial variation along the other. Building a complete hyperspectral image requires an incremental depointing of

the instantaneous field of view (FOV), or a translation of the whole instrument. Typical examples of both applications can be found in Heue et al. (2008) or Lohberger et al. (2004). While the DOAS technique is well validated in terms of accuracy and sensitivity, the need for scanning the scene hampers the detection of dynamic processes. As the scene is sampled slice by slice, the final image does not show a great temporal consistency: different rows (or columns depending on the scanning direction) are temporally disconnected from each other. The time gap can reach several minutes between both edges of the scene.

There are situations where high spatio-temporal resolution is needed. In volcanology for instance, the main measurement technique for the $SO_2$ released in volcanic plumes has recently moved from spectrometers (gas-correlation or DOAS methods) to so-called $SO_2$ cameras (Mori and Burton, 2006; Bluth et al., 2007), although infrared imagers have also been developed (see Platt et al. (2014) for a review of techniques). Their concept is based on taking spectral images of the plume through two interference filters. One filter selects a narrow band of the incident spectrum around 310 nm where $SO_2$ is still strongly

absorbing, the other one captures the light around 330 nm where almost no more absorption takes place. The main advantages are a typical temporal resolution of 1 Hz, the capability to capture dynamic features such as puffs in the plume, and the possibility to determine the plume speed from the sequence of images. The disadvantages are the interference by the plume aerosols caused by the coarse spectral resolution, and the need for regular re-calibration with reference cells filled with $SO_2$ to account for changes of illumination conditions (Kern et al., 2010). More recent concepts now use the combined information of

a spectrometer with the camera spectral images (Lübcke et al., 2013) which yields a greater measurement accuracy.

We present a new instrument, a spectral imager dedicated to measuring the 2-D $NO_2$ field above finite sources like thermal power plants, industrial complexes, cities, volcanoes,... The measurement principle is close to the $SO_2$ camera: snapshots at two wavelengths emphasize the presence of $NO_2$ by taking advantage of absolute differences in the molecule absorption cross section. Contrary to the $SO_2$ cameras which use interference filters, the new instrument relies on an acousto-optical tunable

filter (AOTF) to provide the spectral information. The AOTF can offer sufficient spectral resolution to resolve the structures of the $NO_2$ spectrum. The ability to discriminate between weak and strong absorption within a few nanometres virtually cuts out any sensitivity to aerosol scattering and changes of solar angles. Potential applications tackle urban and industrial pollution monitoring, emission fluxes estimation, satellite-products validation, volcanic plume chemistry.

## 2 Instrument concept

The AOTF-based $NO_2$ camera derives from the ALTIUS instrument (atmospheric limb tracker for the investigation of the upcoming stratosphere). ALTIUS is a space mission project aiming at the retrieval of atmospheric constituents densities with a global geographical coverage and a high vertical resolution. Its primary scientific objective is to measure ozone, but $NO_2$, aerosols, $H_2O$, $CH_4$, polar stratospheric and noctilucent clouds, and other minor species will be measured as well. Measure-





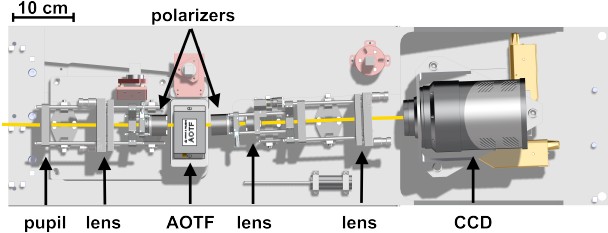

**Figure 1.** Optical layout of the NO2 camera seen from top. Light propagates from left to right through a pupil and a lens doublet, a polarizer selecting vertically-polarized light, the AOTF, a second cross-oriented polarizer, two lens doublets and the detector.

ments will be performed in two different geometries: limb scattering and occultations (Sun, Moon, stars, planets). To address the problem of tangent height registration of previous limb scatter instruments, a spectral imager concept based on a tunable filter has been selected. During the feasibility study, an optical breadboard of the visible (VIS) channel (440-800 nm) was built from commercially-available parts. The detailed description of this breadboard is given in Dekemper et al. (2012). We will

only point out the key features of the concept.

The instrument offers a $6°$ square FOV imaging onto a Princeton Instrument Pixis 512B peltier-cooled CCD detector ($512 \times 512$ pixels). The optical layout (Fig. 1) is linear with an intermediate focal plane located close to the AOTF. To preserve the spectral homogeneity across the image, the design is made telecentric by placing an iris at the object focal point of the first lens. This ensures an identical propagation angle of all light rays through the AOTF.

The most important part of this $NO_2$ camera concept is the AOTF (Chang, 1974). AOTFs have been used in many areas requiring spectral images (agriculture, food industry, fluorescence spectroscopy, etc), but received little attention from the atmospheric remote sensing community. The working principle is based on the interaction of light and sound in a birefringent crystal (see Fig. 2). By the momentum matching of the optical and acoustic waves, a narrow portion of the light spectrum is diffracted into a slightly different direction (a few degrees). If the incident radiation is linearly polarized, the diffracted beam

will leave the crystal with the orthogonal polarization. The spatial and polarimetric dissociations can be combined to achieve very efficient extinction of the unwanted spectrum.

The wave vectors matching condition creates a monotonic relationship between the light wavelength and the sound frequency. The acoustic wave is launched into the crystal by a piezoelectric transducer bonded to one of its facets. Hence, selecting a particular wavelength $\lambda$ simply requires to drive the transducer to the matching frequency $F(\lambda)$. The AOTF spectral

transmission function (STF) closely follows a $\mathrm{sinc}^2$ shape. The amplitude of the STF, which determines the filter diffraction efficiency (DE), is controlled by the acoustic power $P_\mathrm{a}(\lambda)$ which also exhibits a smooth wavelength dependence. The transducer length defines the length of the acousto-optic interaction which directly affects the AOTF bandwidth: a short transducer will induce a larger passband and vice-versa.

The parameters of an AOTF are defined by the crystal elastic and optical properties, and by the propagation directions of

light and sound in the frame of the crystal axes (Voloshinov et al., 2007). The AOTF we used was manufactured out of a



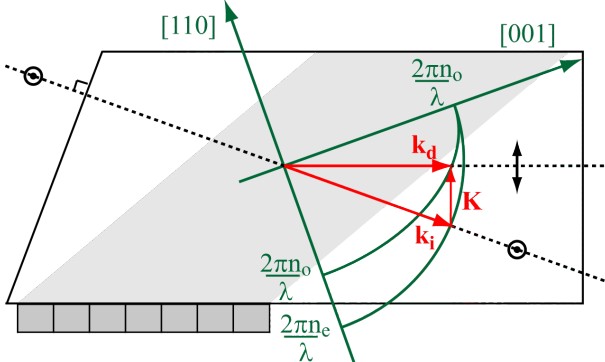

**Figure 2.** Schematics of the acousto-optic interaction in an AOTF (top view). The gray area depicts the acoustic field created by the piezo-electric transducer bonded to a lateral face of the $TeO_2$ crystal. The momentum phase matching of the incident ($\boldsymbol{k_i}$) and diffracted ($\boldsymbol{k_d}$) photons with the acoustic wave ($\boldsymbol{K}$) is represented in the [$\bar{1}10$] crystallographic frame. The phase matching takes advantage of the medium birefringence: incident and diffracted light beams have orthogonal polarisations and different propagation directions which facilitates their selection.

$TeO_2$ crystal by the company Gooch & Housego (U.K.). It offers an aperture of $10 \times 10$ mm$^2$, and a tuning range covering the visible spectrum. Laboratory characterization revealed a transparency better than 90%, and a DE better than 95%. In the relevant spectral range for $NO_2$ measurements, i.e. around 450 nm, the STF showed a bandwidth of 0.6 nm. Typical driving frequencies were around 130 MHz, and less than 100 mW of acoustic power was needed in any circumstances. The theoretical number of resolvable spots at 450 nm is about 350 in the plane of acousto-optic interaction (horizontal axis), and 700 in the vertical direction.

## 3 Measurement principle

There are strong similarities between the measurement principles of a filter-based $SO_2$ camera and an AOTF-based $NO_2$ camera: the FOV needs to be pointed towards the target region (e.g. a plume) while making sure that the background can still be seen in some areas of the image. Two spectral images of the scene are taken: one at a wavelength $\lambda_s$ where there is strong absorption by the target species, another at a wavelength $\lambda_w$ where there is weak absorption. In each image, the signal $S_{ij}(\lambda)$ (in e$^-$) recorded by pixel $ij$ looking at the plume will be normalized by the background signal $S_0(\lambda)$ in order to emphasize the extinction that took place during the crossing of the plume. The optical thickness $\tau_{ij}$ associated with the slant column density (SCD) of the target species observed in the FOV of pixel $ij$ follows from the comparison of the normalized signals recorded at the two wavelengths.

The major difference comes from the capability of the AOTF-based $NO_2$ camera to resolve the fine structures of the absorption cross section $\sigma_{NO_2}$ (Fig. 3). This allows choosing $\lambda_s$ and $\lambda_w$ very close to each other (a few nm), minimizing the interference by broadband absorbing and scattering species like aerosols.


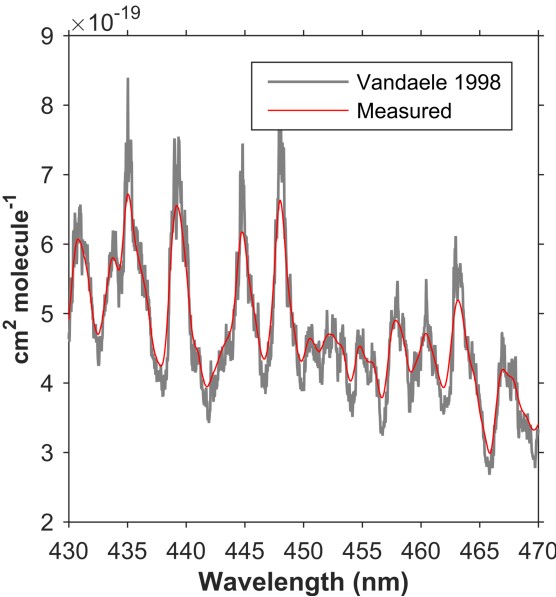

**Figure 3.** NO$_2$ absorption cross section measured with a fourier transform spectrometer (gray line, (Vandaele et al., 1998)), and with this NO$_2$ camera in the laboratory (red line). At 450 nm, the spectral resolution of both datasets are 0.04 nm and 0.6 nm respectively.

## 3.1 Mathematical model

As AOTFs do not treat different polarizations identically, an AOTF-based NO$_2$ camera exhibits a strong polarization sensitivity. The polarization state of a stream of light is described by the Stokes vector $\boldsymbol{s} = (I, Q, U, V)^T$, where $I = I^h + I^v$ and $Q = I^h - I^v$ with $I^h$ and $I^v$ being the light intensity along the horizontal and vertical axes of a scene frame. $U$ and $V$ also refer to

5  the orientation of the polarization ellipse but they will not be discussed further because they do not participate if the AOTF and its surrounding polarizers are well aligned.

When light passes through a polarizing part, its Stokes vector can be changed. A polarizing element is therefore represented by a $4 \times 4$ transfer matrix: the Mueller matrix $\boldsymbol{M}$. A chain of optical elements is represented by the product of their Mueller matrices. In our design, the light passes first through a vertical linear polarizer, then the AOTF, and finally a horizontal linear

10  polarizer. The Stokes vector representing the light leaving the second polarizer is therefore given by $\boldsymbol{s'} = \boldsymbol{M}_{\mathrm{Ph}}.\boldsymbol{M}_{\mathrm{AOTF}}.\boldsymbol{M}_{\mathrm{Pv}}.\boldsymbol{s}$. The Mueller matrices of the elements are as follows:

$$\boldsymbol{M}_{\mathrm{AOTF}} = \frac{A}{2} \begin{pmatrix} 1 & -1 & 0 & 0 \\ 1 & -1 & 0 & 0 \\ 0 & 0 & 0 & 0 \\ 0 & 0 & 0 & 0 \end{pmatrix}, \tag{1}$$



$$M_{\mathrm{Pv}} = \frac{1}{2}\begin{pmatrix} \eta_e^2 + \eta_t^2 & \eta_e^2 - \eta_t^2 & 0 & 0 \\ \eta_e^2 - \eta_t^2 & \eta_e^2 + \eta_t^2 & 0 & 0 \\ 0 & 0 & 2\eta_e\eta_t & 0 \\ 0 & 0 & 0 & 2\eta_e\eta_t \end{pmatrix}, \quad M_{\mathrm{Ph}} = \frac{1}{2}\begin{pmatrix} \eta_t^2 + \eta_e^2 & \eta_t^2 - \eta_e^2 & 0 & 0 \\ \eta_t^2 - \eta_e^2 & \eta_t^2 + \eta_e^2 & 0 & 0 \\ 0 & 0 & 2\eta_e\eta_t & 0 \\ 0 & 0 & 0 & 2\eta_e\eta_t \end{pmatrix}, \tag{2}$$

where $A$ is the amplitude of the AOTF STF (i.e. its DE, $0 \leq A \leq 1$), $\eta_t^2$ is the attenuation of the light intensity along the polarizer transmission axis, and $\eta_e^2$ does the same for the extinction axis. Assuming that all three elements have their transmission and extinction axes well aligned, the total Mueller matrix of the camera is simply $M = \eta_t^4.M_{\mathrm{AOTF}}$. As the detector only measures the total light intensity, the first element of the Stokes vector is the only meaningful quantity: $s'(1) = A.\eta_t^4.(I - Q)/2 = A.\eta_t^4.I^v$. Hence, in the present configuration, the $NO_2$ camera is only sensitive to vertically polarized light.

We now have a description of the light intensity which will be measured by the detector, but we still have to account for the extinction by the lenses ($T$) and the quantum efficiency (QE) of the detector. These terms exhibit a smooth wavelength dependence. For the AOTF STF, one can use $\mathcal{F}(\lambda; \lambda_c) = A(\lambda_c).G(\lambda - \lambda_c)$, where $G$ is essentially a $\mathrm{sinc}^2$ function. Moreover, some parameters are susceptible to vary across the FOV, yielding a pixel-to-pixel variation. This is particularly true when image planes are located close to optical surfaces (mainly the AOTF and the detector). Finally, the electronic current (in e$^-$ s$^{-1}$) found in pixel $ij$ when the AOTF is tuned to $\lambda_c$ is given by:

$$C_{ij}(\lambda_c) = \int A_{ij}(\lambda_c).\eta_t^4(\lambda).I_{ij}^v(\lambda).G(\lambda - \lambda_c).T(\lambda).\mathrm{QE}_{ij}(\lambda)\,d\lambda,$$
$$= A_{ij}(\lambda_c).\eta_t^4(\lambda_c).T(\lambda_c).\mathrm{QE}_{ij}(\lambda_c)\int I_{ij}^v(\lambda).G(\lambda - \lambda_c)\,d\lambda,$$
$$= r_{ij}(\lambda_c)\int I_{ij}^v(\lambda).G(\lambda - \lambda_c)\,d\lambda. \tag{3}$$

The decision to leave the smoothly varying parameters out of the integral is supported by the narrow passband of the AOTF (0.6 nm). Their product forms the instrument response at pixel $ij$ and wavelength $\lambda_c$: $r_{ij}(\lambda_c)$. The remaining integral is simply the convolution of the vertically-polarized incident light intensity with the AOTF STF.

Suppose now that pixel $ij$ is looking through an optically thin plume. $NO_2$ and other species will absorb or scatter photons and decrease the background light intensity $I_0^v$ according to the Beer-Lambert law of extinction:

$$I_{ij}^v(\lambda) = I_0^v(\lambda).\exp\left(-\tau_{\mathrm{NO_2}\,ij}(\lambda) - \tau_{\star\,ij}(\lambda)\right), \tag{4}$$

where $\tau_{\mathrm{NO_2}\,ij}$ denotes the plume optical thickness caused by absorption by $NO_2$ along the light path reaching of pixel $ij$, and $\tau_{\star\,ij}$ is the effective optical thickness of all other chemical species and particles. Over the passband of the AOTF, one can consider $\tau_\star(\lambda)$ as a constant value $\tau_\star(\lambda_c)$ and replace $\tau_{\mathrm{NO_2}}(\lambda)$ by its weighted average:

$$\overline{\tau}_{\mathrm{NO_2}}(\lambda_c) = \frac{\int \tau_{\mathrm{NO_2}}(\lambda).G(\lambda - \lambda_c)d\lambda}{\int G(\lambda - \lambda_c)d\lambda}.$$

As the optical thickness is defined by the product of the trace gas SCD $k$ with its absorption cross-section $\sigma$, it is clear that $\overline{\tau}_{\mathrm{NO_2}}(\lambda_c) = k_{\mathrm{NO_2}}.\overline{\sigma}_{\mathrm{NO_2}}(\lambda_c)$. Under these assumptions, one can replace Eq. (4) into Eq. (3) and write for the pixel photoelectric



current:

$$C_{ij}(\lambda_c) = r_{ij}(\lambda_c) . \exp\left(-\overline{\tau}_{\text{NO}_2\,ij}(\lambda_c) - \tau_{\star\,ij}(\lambda_c)\right) . \int I_0^v(\lambda) . G(\lambda - \lambda_c)\,\mathrm{d}\lambda. \tag{5}$$

In the meantime, other pixels have been looking at the unattenuated background intensity $I_0$. Suppose that one of them is pixel $mn$. According to Eq. (3), we have:

$$C_{mn}(\lambda_c) = r_{mn}(\lambda_c) . \int I_0^v(\lambda) . G(\lambda - \lambda_c)\,\mathrm{d}\lambda.$$

Averaging all these background-looking pixels yields the reference current associated with the background intensity:

$$C_0(\lambda_c) = r(\lambda_c) . \int I_0^v(\lambda) . G(\lambda - \lambda_c)\,\mathrm{d}\lambda, \tag{6}$$

with $r$ representing the average instrument response. Dividing $C_{ij}$ by $C_0$ yields the transmittance of the plume only:

$$T_{ij}(\lambda_c) = \frac{\frac{C_{ij}(\lambda_c)}{r_{ij}(\lambda_c)}}{\frac{C_0(\lambda_c)}{r(\lambda_c)}} = \exp\left(-\overline{\tau}_{\text{NO}_2\,ij}(\lambda_c) - \tau_{\star\,ij}(\lambda_c)\right). \tag{7}$$

If the plume transmittance is calculated for each of the two carefully-selected wavelengths $\lambda_w$ and $\lambda_s$, then it is possible to cancel the interference of other species (because $\tau_{\star}(\lambda_w) = \tau_{\star}(\lambda_s)$ if $\lambda_w$ and $\lambda_s$ are very close). Introducing the relative instrument response at pixel $ij$: $\rho_{ij}(\lambda) = r_{ij}(\lambda)/r(\lambda)$, the ratio of transmittances

$$\frac{T_{ij}(\lambda_w)}{T_{ij}(\lambda_s)} = \frac{\frac{C_{ij}(\lambda_w)}{C_0(\lambda_w)\rho_{ij}(\lambda_w)}}{\frac{C_{ij}(\lambda_s)}{C_0(\lambda_s)\rho_{ij}(\lambda_s)}} = \exp\left(\overline{\tau}_{\text{NO}_2\,ij}(\lambda_s) - \overline{\tau}_{\text{NO}_2\,ij}(\lambda_w)\right), \tag{8}$$

which yields the $\text{NO}_2$ SCD observed by pixel $ij$:

$$k_{\text{NO}_2\,ij} = \frac{1}{\overline{\sigma}_{\text{NO}_2}(\lambda_s) - \overline{\sigma}_{\text{NO}_2}(\lambda_w)} . \ln\left(\frac{T_{ij}(\lambda_w)}{T_{ij}(\lambda_s)}\right). \tag{9}$$

## 3.2  Ancillary data

Equation (9) shows that the $\text{NO}_2$ SCD can be obtained from a combination of measurements (the detector signal), cross-section data and the knowledge of the instrument response. In the results presented below, the cross-section is taken from Vandaele et al. (1998). For the $\rho_{ij}$ coefficients, an *ad hoc* method was set up to build a synthetic flat field. Taking advantage of a cloudy weather (100% cloudiness), long-exposure frames (10 s) were captured at the required wavelengths looking at zenith. The mean image obtained from tens of such frames constitutes the instrument response to a synthetic, radiometrically flat, scene. This allows to remove wavelength-dependent non-uniformities which can be relatively pronounced in the AOTF for instance.

Access to the photoelectric current strictly proportional to the signal (i.e. $C_{ij}$ and $C_0$) implies that voltage offset, dark current and stray light have been subtracted from the raw data. In this respect, AOTFs offer a unique feature: one can turn them off. This is simply done by setting the RF level to 0. An image acquired in these conditions contains anything but the real signal (i.e. dark current, offset, straylight). Using $D_{ij}$ and $D_{ij}^{\text{off}}$ to represent the raw signal of pixel $ij$ (in digital numbers, DN) when the AOTF is turned on or off respectively, the photo-electric signal is given by

$$S_{ij} = \frac{D_{ij} - D_{ij}^{\text{off}}}{G}, \tag{10}$$



where $G$ is the sensor gain (in DN/e$^-$). The only precaution is to take these dark images regularly because, the straylight being connected with the general illumination conditions (e.g. solar angles), it will vary with time.

### 3.3 Data averaging and multiple image doublets

It is often necessary to repeat the measurements in order to average out transient features and increase the signal-to-noise ratio (SNR). Assuming that only the plume optical transmission varies, we can write a time-dependent version of Eq. (5):

$$C_{ij}(\lambda_c, t) = r_{ij}(\lambda_c) . \exp\left(-\overline{\tau}_{\mathrm{NO}_2\ ij}(\lambda_c, t) - \tau_{\star\ ij}(\lambda_c, t)\right) . \int I_0^v(\lambda) . G(\lambda - \lambda_c)\, \mathrm{d}\lambda. \tag{11}$$

The time-averaged optical thickness $\tau(\lambda, \overline{t})$ can be obtained from the geometric mean of the consecutive images:

$$\sqrt[N]{\prod_{k=1}^N C_{ij}(\lambda_c, t_k)} = r_{ij}(\lambda_c) . \exp\left(-\frac{1}{N}\sum_{k=1}^N \overline{\tau}_{\mathrm{NO}_2\ ij}(\lambda_c, t_k) + \tau_{\star\ ij}(\lambda_c, t_k)\right) . \int I_0^v(\lambda) . G(\lambda - \lambda_c)\, \mathrm{d}\lambda.$$

$$= r_{ij}(\lambda_c) . \exp\left(-\overline{\tau}_{\mathrm{NO}_2\ ij}(\lambda_c, \overline{t}) - \tau_{\star\ ij}(\lambda_c, \overline{t})\right) . \int I_0^v(\lambda) . G(\lambda - \lambda_c)\, \mathrm{d}\lambda. \tag{12}$$

Another means of increasing the reliability of the measurements is to use different doublets, i.e. pairs of $\lambda_w$ and $\lambda_s$. If the transmittance is known for several doublets, their product strengthens the $\mathrm{NO}_2$ SCD retrieval by providing information from multiple spectral regions. If $\Delta\sigma_{\mathrm{NO}_2} = \overline{\sigma}_{\mathrm{NO}_2}(\lambda_s) - \overline{\sigma}_{\mathrm{NO}_2}(\lambda_w)$, then for two doublets, we have for the SCD:

$$k_{\mathrm{NO}_2\ ij} = \frac{1}{\Delta\sigma_{\mathrm{NO}_2}^{(1)} + \Delta\sigma_{\mathrm{NO}_2}^{(2)}} . \ln\left(\frac{T_{ij}(\lambda_{w1}) . T_{ij}(\lambda_{w2})}{T_{ij}(\lambda_{s1}) . T_{ij}(\lambda_{s2})}\right). \tag{13}$$

This approach can potentially attenuate a bias in one of the measurements.

### 3.4 Error budget and instrument sensitivity

Obviously, one can work out Eq. (5) with the classical first-order Taylor expansion approximation to determine the uncertainty on the $\mathrm{NO}_2$ SCD. This approach will require estimates of the uncertainty on the photon counts $C_{ij}$, on the background signal $C_0$, on the relative instrument response $\rho_{ij}$, and on the cross-section data $\sigma_{\mathrm{NO}_2}$. These estimates are not always easily obtained, and we briefly discuss each of them.

The photo-electric counting rates $C_{ij}$ are obtained from Eq. (10): $C_{ij} = S_{ij}/t$, where $t$ is the sensor exposure time. It is reasonable to assume that the camera operator selected acquisition settings ensuring that the signal is well into the shot noise regime: $\sigma_{C_{ij}} = \sqrt{S_{ij}}/t$. With signals exceeding $10^4$ e$^-$ in 1 second (the case in the examples below), the relative uncertainty on $C_{ij}$ will be below 1%.

The background signal $C_0$ is estimated by averaging the pixels looking at the background of the scene. While one would presume that the averaging of a large number of such pixels should yield a very high precision, the efficiency is limited by the difficulty to identify pixels effectively looking at the background. Automated data processing needs a screening of each image to determine if a pixel is looking at the plume, the background, a cloud, or even a bird. This screening is based on the interpretation of the raw signals and, for instance, it sometimes fails to recognize pixels which still have in their FOV the





residual $NO_2$ molecules left by a past position of the plume. From our experience, the relative uncertainty on $C_0$ determined from a single image is generally larger than 1% (determined from the sample standard deviation). Using multiple images as explained in section 3.3 reduces this uncertainty, as $C_0$ is computed for each image then averaged. A 1% total relative error is achievable with a few images.

The relative instrument response non-uniformity $\rho_{ij}$ can be obtained from a homogeneous scene (i.e. a flat field such that $I_{ij}(\lambda) = I(\lambda) \forall i,j$). In this particular case, $\rho_{ij}(\lambda) = C_{ij}(\lambda)/C(\lambda)$, where $C(\lambda)$ is the average of $C_{ij}$ over a large number of pixels. If the flat field is built from a number of relatively homogeneous images under the assumption that their average is truly flat, then the uncertainty on the flatness participates to the error budget of $\rho_{ij}(\lambda)$ and quickly becomes the driver (signal shot noise is surpassed). This error source is a generic problem of all imaging systems but remains difficult to quantify. The only

certainty is that it drops with the sample size.

The $NO_2$ absorption cross-section data are taken from Vandaele et al. (1998) who report a total relative uncertainty of 3% at a resolution of 2 cm$^{-1}$ (0.04 nm at 450 nm). Taking our coarser resolution into account (about 0.6 nm), the uncertainty drops to about 0.8%. However, the AOTF tuning curve is temperature-dependent, with a typical drift of +0.1 nm per Kelvin (Ohmachi and Uchida, 1970; Uchida, 1971). The driving electronics is currently not enslaved to a temperature sensor. The exact

measurement wavelength is computed at the processing stage. Depending on the amount of wavelength drift, the uncertainty on $\sigma_{NO_2}(\lambda)$ can reach 5-10%.

The minimum relative uncertainty on the $NO_2$ SCD will be reached if the uncertainty on the plume transmittance $T$ is driven by $C_0$. Assuming $\sigma_T/T = 1\%$, and taking into account a 5% error on the cross-section term (with a typical value for $\sigma_{NO_2}(\lambda_s) - \sigma_{NO_2}(\lambda_w) = 2 \times 10^{-19}$), one obtains $\sigma_k = 5 \times 10^{16}$ molecules cm$^{-2}$. If one assumes less favorable conditions like

a 1% uncertainty on $\rho$ yielding $\sigma_T/T = 2\%$ and a 10% error on the cross-section, then the SCD error reaches $10^{17}$ molecules cm$^{-2}$.

## 4 Application to the remote sensing of $NO_2$ at a coal-fired power plant

The data of a spectral imager such as the $NO_2$ camera are more easily exploited if a number of observational requirements are satisfied. First, the camera must be placed at a location where both the plume and the background can be captured within

the same image. Second, the target plume must remain optically thin in order to preserve the assumption of the Beer-Lambert extinction along a straight light path. Finally, scattered clouds behind the plume will corrupt the retrieval and should be avoided.

These three requirements were sometimes fulfilled during the second Airborne ROmanian Measurements of Aerosols and Trace gases (AROMAT-2) campaign to which we participated in August 2015. The campaign aimed at joining the efforts of several European research institutes and universities to spatially and temporally characterize the emissions from two types of

site: a large city (Bucharest), and point sources (large thermal power plants in the Jiu Valley, Romania). Both sites should eventually serve as validation targets for the ESA TROPOMI/S-5P mission.

The $NO_2$ camera was placed at a distance of 2.5km from a group of 4 stacks belonging to Turceni's power plant, the largest in Romania (330MW per turbine, 2000GWh/year total electric power generation of which more than 93% out of coal). Figure





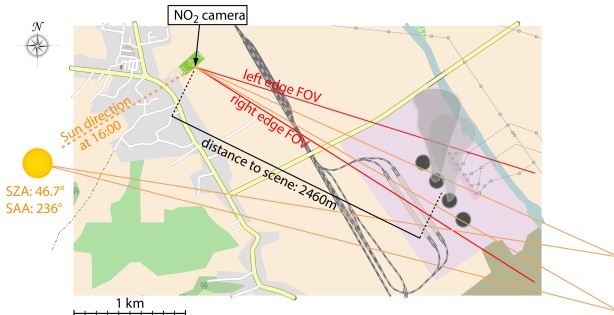

**Figure 4.** Observational geometry during the AROMAT-2 campaign at Turceni's power plant. The $NO_2$ camera was installed on a football pitch looking at the four 280m-tall stacks. The red lines delimit the camera horizontal FOV ($6°$). The direction of the Sun at 16:00 local time is approximately indicated, together with two rays illustrating the scattering behind the scene towards the camera. One of the rays passes through the plume, the other one passes by. [Map data from OpenStreetMap]

depicts the measurement geometry. Our location was 44.6792°N 23.3788°E, the line of sight (LOS) azimuth angle ranged from 113° (left edge of the image) to 119° (right edge) Eastward from North, and the LOS zenith angle ranged between 75.5° (top edge) and 81.5° (bottom edge). We only report on measurements performed on the 24th between 16:15 and 16:30 as the observational conditions were close to ideal and better illustrate the performance of the instrument.

## 4.1 Exhaust plume $NO_2$ SCD field

As explained in section 3.1, the 2-D $NO_2$ SCD field is computed from at least two spectral images recorded at wavelengths showing a significant difference of absorption cross-section. To increase the reliability of the measurements, 4 doublets of wavelengths were used: $\lambda_{w1} = 441.8$ and $\lambda_{s1} = 439.3$, $\lambda_{w2} = 446.7$ and $\lambda_{s2} = 448.1$, $\lambda_{w3} = 437.9$ and $\lambda_{s3} = 435.1$, $\lambda_{w4} = 465.8$ and $\lambda_{s4} = 463.2$. The automated acquisition system was in charge of synchronizing the driving of the AOTF with the image acquisition. A nominal acquisition sequence started by setting the appropriate RF signal for the AOTF to filter at $\lambda_{w1}$, open the CCD shutter for 0.5s, readout the image and repeat these operations for the 7 other wavelengths. After completion of the nominal sequence, a picture with the AOTF turned off is taken and the nominal sequence is resumed. The dwell time between the closing of the shutter and its re-opening was 1.3 s. In the plane of the stacks, the image footprint spans an area of $250 \times 250$ m$^2$ with a 50 cm sampling.

The data analysis revealed that the images from the 2nd and 4th doublets were the less noisy, because of a larger natural radiance and sensor sensitivity compared to the wavelengths of doublets 1 and 3. Also, due to the plume displacement over time (wind), and the presence of moving and changing inhomogeneities across the plume (puffs, turbulent eddies), it was necessary to perform time averaging (section 3.3). Indeed, the 1.3 second between two consecutive images is already a long time for features moving at a typical 5 m s$^{-1}$ speed (corresponding to 10 pixels per second).

Figure 5 shows the $NO_2$ SCD field retrieved from the averaging of images taken at $\lambda_{w2}$, $\lambda_{s2}$, $\lambda_{w4}$ and $\lambda_{s4}$ (12 of each) using the method described in Section 3.3. For each wavelength, the background signal $C_0$ was determined from image areas





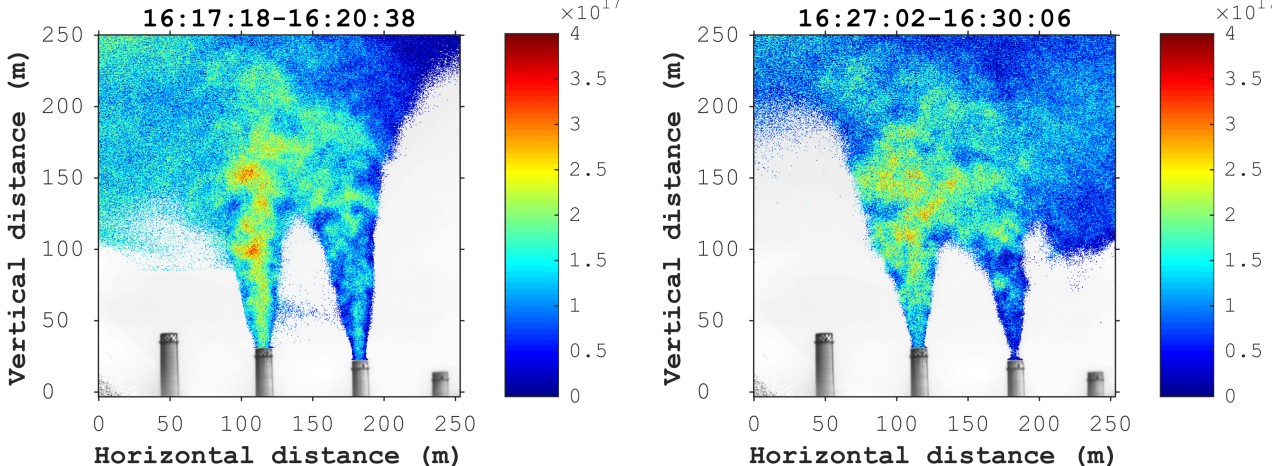

**Figure 5.** Sample $NO_2$ SCD field obtained from the averaging of images acquired at $\lambda_{w2} = 446.7$nm, $\lambda_{s2} = 448.1$nm, $\lambda_{w4} = 465.8$nm and $\lambda_{s4} = 463.2$nm (12 of each). The colorscale shows the plume $NO_2$ SCD in molecules $cm^{-2}$. The x- and y-axis show the image dimensions in the scene plane, while the title gives the time span (local time).

*untouched* by the plume. The relative error is about 0.5% (estimated from the standard deviation $\sigma_{C_0}$ of the pixels sample yielding $C_0$). Within this precision, no variation of $C_0$ across the FOV could be significantly detected. The reason is the relatively small FOV of the camera (about 6°) combined with a high Sun at the time of the measurements (making the scene illumination quite homogeneous). In Fig. 5, the background grayscale image is the mean image at $\lambda_{w4}$, whereas the pixels

where the SCD is computed were selected based on the criterium $C_{ij} < C_0 - 2\sigma_{C_0}$. Investigating the random fluctuations observed in various areas of the SCD field, one can estimate the detection limit to about $5 \times 10^{16}$ molecules $cm^{-2}$.

## 4.2   $NO_2$ emission fluxes and synergies with $SO_2$ cameras

The capability of resolving the $NO_2$ SCD field with a high spatial and temporal resolution provides new possibilities for the understanding of the plume chemistry. Coal combustion processes yielding the formation of nitrogen and sulfur species are

well-known (Flagan and Seinfeld, 1988), and several reactive plume models can simulate the transport, formation and removal of these species over different scales. These models are generally validated by in-situ air sampling at distances of several kilometres downwind (see for instance Chowdhury et al. (2015)). Very few experiments attempted to characterize the reactive content of the early plume, where the reactions are still governed by the combustion products (Hewitt, 2001). In most cases, a DOAS scanning system was used (Lee et al., 2014, 2009; Lohberger et al., 2004). The same technique was also used for $SO_2$,

but to a lesser extent since the introduction of filter-based $SO_2$ cameras (Smekens et al., 2015). Recently, imaging Frourier-transform spectroscopy (IFTS) demonstrated capability for the measurement of a number of mid-infrared emitting species such as $CO_2$ and $SO_2$ (Gross et al., 2010). However, NO but not $NO_2$, can be retrieved with this technique.





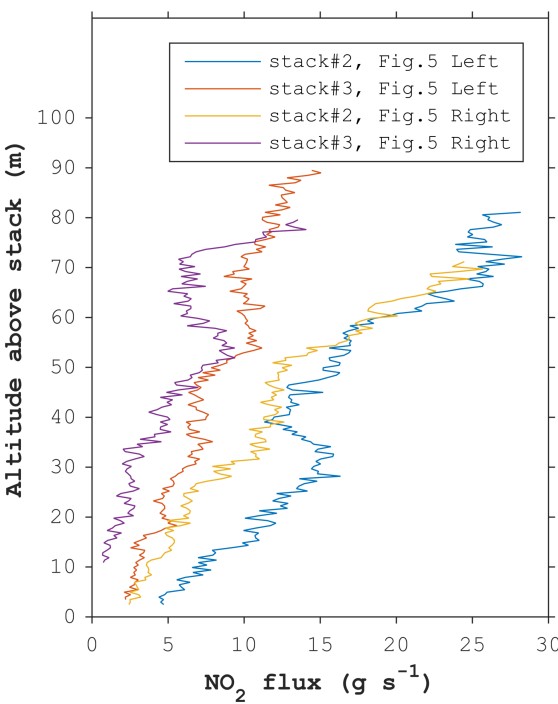

**Figure 6.** $NO_2$ flux computed through the plume horizontal cross-section as a function of altitude. Stacks height is 280m. A symmetric Gaussian dispersion is assumed up to the region of apparent intersection of the two plumes.

An undisputed advantage of imaging systems with high temporal resolution is their ability to track the displacement of remarkable features from one image to another. We used the complete time series of spectral images (50 sequences of 8 spectral images at a rate of 0.5 Hz) to determine the vertical speed of the plume. This was done by tracking signal features created by local increase or decrease of the $NO_2$ concentration. On average, a vertical speed of $4.8 \pm 0.5$ m s$^{-1}$ was observed.

5   Furthermore, assuming a Gaussian dispersion of the plume, one can infer a circular cross-section from the apparent width of the plume at each detector row (i.e. every 50 cm above the stack outlet). As a result, a profile of emission flux (in g s$^{-1}$) can be drawn. Figure 6 shows the $NO_2$ emission flux as a function of altitude up to a height above which the two plumes cannot be discriminated anymore. The fluxes were calculated from the two SCD maps of Fig. 5, and both stacks. The increase is the result of the conversion of NO into $NO_2$ mainly by the reactions $2NO + O_2 \rightarrow 2NO_2$, and $NO + HO_2 \rightarrow NO_2 + OH$ (Flagan

10   and Seinfeld, 1988; Miller and Bowman, 1989), even if these processes are balanced by the photodissociation of $NO_2$ as soon as it reaches open air under day light ($NO_2 + h\nu \rightarrow NO + O$). Qualitatively, these results agree well with the increase reported by Lee et al. (2009) in a study of the rate of increase of $NO_2$ above power plant stacks. The analysis of Fig. 6 reveals that within the method approximations, the $NO_2$ concentration in the plume increases at a rate ranging from 0.75 to 1.25 g s$^{-1}$ ($9.8 \times 10^{21}$-$1.6 \times 10^{22}$ molecules s$^{-1}$) on average for the first 20 seconds.



The knowledge of the spatial distribution of $NO_2$ can also prove useful to correct measurements marked by interference from $NO_2$. A good example is with $SO_2$ cameras where the $SO_2$ SCD field is retrieved by comparing the plume transmittance around 310 and 330 nm. In this range, $NO_2$ is also absorbing and its cross-section roughly doubles from 310 to 330 nm. Therefore, if both molecules are present in the plume, the $SO_2$ camera alone cannot disentangle their respective signature. So far, this interference has been overlooked in $SO_2$ camera validation exercises (Smekens et al., 2015; Kern et al., 2010). In the case of the plumes shown in Fig. 5 for instance, a $SO_2$ camera such as the one used by Smekens et al. (2015) would observe a $\Delta\tau_{NO_2} = 0.04$ when the $NO_2$ SCD reaches $3 \times 10^{17}$ molecules cm$^{-2}$. This variation of optical thickness corresponds to a $SO_2$ SCD of about $1.6 \times 10^{17}$ molecules cm$^{-2}$, which is twice the detection limit reported in Smekens et al. (2015). Clearly, the bias would increase with higher concentrations of $NO_2$. Taking advantage of the similar spatial resolution of both instruments, the $NO_2$ camera can provide a complete correction map for the $SO_2$ data.

## 5 Conclusions

We have described a new passive atmospheric remote sensing instrument for the measurement of $NO_2$ slant column densities (SCDs) above strong sources. It is based on an acousto-optical tunable filter (AOTF) which offers a sufficient acceptance angle to be placed in an imaging system, and the necessary resolution for taking advantage of the fine structures of the $NO_2$ absorption cross-section. The AOTF is electrically driven, such that fast synchronized acquisitions of spectral images are possible.

The measurement principle is similar to the filter-based $SO_2$ camera: SCDs are retrieved from at least two spectral images where absorption by the target molecule is significantly different. Wavelengths are picked in the range 440-470 nm. Thanks to its higher spectral resolution, the AOTF-based $NO_2$ camera can perform its measurements within a few nanometres. This makes the sensitivity to aerosols negligibly small.

A mathematical frame has been developed, and the different sources of error have been addressed. In applications focusing on relatively high spatio-temporal resolution, the $NO_2$ SCD detection limit is about $5 \times 10^{16}$ molecules cm$^{-2}$. Different measurement geometries offering longer staring time or more stable targets would yield a lower limit.

The $NO_2$ camera was successfully tested during the AROMAT-2 campaign where measurements of $NO_2$ SCD fields above the flue gas stacks of a coal-fired power plant were performed with a temporal resolution of 3 minutes and a spatial sampling of 50 cm (for a complete scene of $250 \times 250$ m$^2$). Values up to $4 \times 10^{17}$ molecules cm$^{-2}$ were observed. The quality of the data allowed to clearly identify the conversion process from NO to $NO_2$ in the early plume, providing quantitative information on the plume dynamic chemistry. In another example of application, the measurements were used to show how the knowledge of the high resolution $NO_2$ field can help to correct $SO_2$ camera data. If overlooked, the interfering absorption of $NO_2$ can yield a significant bias in the retrieved $SO_2$ SCDs. Other applications range from emission monitoring to volcanic plume chemistry.

While the concept is mature, a number of improvement directions are still being investigated. The most promising ones are the implementation of a temperature feedback loop to reduce the uncertainty on the filtered wavelength, and the replacement of the CCD by a CMOS in order to reduce the cooling needs and increase the temporal resolution of the measurements.





*Author contributions.* E. Dekemper developed the $NO_2$ camera measurement principle, led the characterization and the participation to the AROMAT-2 campaign, and processed the data. B. Van Opstal and J. Vanhamel developed the acquisition software and the AOTF driving electronics, and participated to the campaign. D. Fussen is at the origin of the instrument concept and supported its development in the frame of the ALTIUS project. The authors declare that they have no conflict of interest.

5    *Acknowledgements.* This work was funded under PRODEX contract 4000110400. Participation to the AROMAT-2 campaign was funded under ESA contract 4000113511. The authors would like to thank Alexis Merlaud for inviting them to participate to the AROMAT-2 campaign. E. Dekemper would like to thank Kerstin Stebel for the interesting discussions on $SO_2$ cameras, and Vitaly Voloshinov for his support in all AOTF-related matters.





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
