# Peer review of "The AOTF-based NO2 camera"

_Atmospheric Measurement Techniques, 2016_

## Referee Comment (RC1) · C. Kern (Referee) · 23 Sep 2016

General Comments

This manuscript describes the development of an optical device specifically for imaging nitrogen dioxide (NO2) gas plumes. The NO2 camera measures incident scattered solar radiation in the visible spectral (blue) region. An acousto-optical tunable filter (AOTF) is used to isolate a relatively narrow (0.6 nm) spectral window, only allowing radiation with wavelengths within this bandpass onto the detector. The central wavelength of the bandpass is adjusted by setting the driving piezoelectric transducer to the matching frequency. This novel design is highly innovative, and the manuscript is well-written. I recommend publishing this paper in Atmospheric Measurement Techniques and have only a few, relatively minor comments.

Specific Comments

P2, L3 – Perhaps also cite Platt et al 1979, as to my knowledge, this was the first DOAS measurement of $NO_2$. Platt U, Perner D, Patz HW. Simultaneous Measurement of Atmospheric $CH_2O$, 03, and $NO_2$ by Differential Optical Absorption. J Geophys Res. 1979;84(C10):6329-6335.

P2 L9 – Here you discuss the time resolution of imaging DOAS instruments vs that of the AOTF-based camera. It occurs to me that it could be useful to mention some relevant physical considerations here. Both instruments can only collect radiation that is available, i.e. solar scattered radiation. Both use similar detectors, so there is no advantage of one over the other in terms of being able to detect available light. While the imaging DOAS only records one line (or, in the case of whisk-broom imaging, one pixel) at a time, it does record a large number of different wavelengths (typically 1,000 or more) coincidentally. The AOTF camera does the opposite – it records the full image, but only 1 wavelength at a time. So the question is whether there really is a physical advantage of one technique over the other. I guess one could argue that not all the wavelengths measured by the imaging DOAS are necessary, but that depends on the application and can actually be adapted as wanted and e.g. tuned to a very specific wavelength range if only interested in $NO_2$. And because so many independent wavelengths are measured, the signal to noise ratio for a single acquisition is significantly better than if only 1 pair of wavelengths is measured (as you mention and show later on in the paper). So I'm curious whether there actually is a physical advantage of one technique over the other and if so, where does it come from? Or is it just that the camera is particularly well-suited to measuring $NO_2$, while the imaging DOAS instruments that have been built in the past have not been optimized just for $NO_2$?

P2 L11 – I disagree that the main measurement technique for quantifying volcanic $SO_2$ has moved to $SO_2$ cameras, and I don't think this will happen in the near future. There are many more scanning DOAS instruments still in use than there are cameras, and the DOAS have some very important advantages too (e.g. being able to measure other gases like BrO and OClO, having a slightly better detection limit, being essentially calibration-free, and having the ability to correct for some scattering effects using additional spectral information that is available).

P3 L9 – I may be wrong, but I don't think it's physically possible to design an optical system such that all rays propagate through the AOTF exactly in parallel. Are you sure this is true? I guess a pupil will reduce beam divergence, but will also reduce the amount of light entering the system. Is there some optimal configuration?

P3 L21 – I don't understand why the acoustic power depends on the wavelength. Can you please clarify?

P4 L5 –Does this mean that you can resolve 350 independent wavelengths and 700 spatial pixels? Can you explain where these numbers come from? I'm not quite following.

P7 L7 – Can you explain a little bit what factors are relevant for 'carefully selecting' the two wavelengths for the measurement? I guess the spectral radiance of the incident scattered radiation and the differential optical depth of the absorption cross-section features play a role.

P9 L14 – Please clarify what you mean with 'not enslaved'. Does this mean that the temperature is not monitored? Or is the instrument not temperature-stabilized? It occurs to me that the errors associated with temperature drift of the bandpass center wavelength could be considerable greater than 5 to 10% if the temperature is not stabilized. For example, if the temperature changes by 5K, thus causing a shift in transmittance wavelength of 0.5nm, then the differential absorption cross section at this wavelength can change by approximately 50%. (e.g. looking at the difference between differential absorption at 448.2 and 448.7nm. In other words, instead of centering the bandpass on an absorption band, the bandpass would move to about the center between an absorption band and the next transmission band, reducing the measured absorption signal by about half!

P11 L1- Can you please clarify what you mean by ‘relative error’? Is this the precision of an individual pixel? I’m somewhat surprised that the value is so low, as the pixel-to-pixel ‘noise’ in SCD values seems to be about 5e16 molec/cm2, which would correspond to about 12%. I think this is the number you mention below. Or is the 0.5% the accuracy? But that seems very ambitious too, based on your previous discussion of errors associated with different sources.

P12 – Can you comment on whether the plumes from the stacks were visible, i.e. did they contain condensed water vapor or were the translucent. Perhaps add a photograph to figure 5? If the plumes did contain water droplets close to the stack, they may not have been optically thin at this point. In other words, the aerosols could influence the light paths through the plume. This could lead to an underestimation of gas concentration in the plumes and may influence the conversion rate estimates calculated from this data. See Kern et al 2013 for a description of the effect on SO2 cameras, though your NO2 camera will react somewhat differently. Kern C, Werner C, Elias T, Sutton AJ, Lübcke P. Applying UV cameras for SO2 detection to distant or optically thick volcanic plumes. J Volcanol Geotherm Res. 2013;262:80-89. doi:10.1016/j.jvolgeores.2013.06.009.

Technical Corrections

P2 L27 – Potential applications INCLUDE urban and industrial pollution. . .

P2 L31 – ALTIUS is a space mission project AIMED at the retrieval of atmospheric COMPOSITION with . . .

P3 L3 – I think you mean that an optical PROTOTYPE of the visible channel was built on a BREADBOARD.

P3 L6 – The instrument IMAGES a 6 degree field of view onto a . . .

P4 L12 - . . . in order to QUANTIFY the extinction. . .

P5 Figure 3 caption – consider replacing MEASURED with AOTF, since both crosssections were actually measured, just with different instruments.

P6 L3 – and . . . IS THE attenuation along the extinction axis.

P6 L8 – I believe T is the transmittance of the lenses, not the extinction

P6 Equation 3 – The second '=' sign should probably be replaced by an 'approximately equals' sign because this step is an approximation.

P6 L21 – Please give this equation a number too.

P6 L 23 – Under these assumptions, one can INSERT Eq. 4 . . .

P7 L13 – I'm not sure that the coefficients rho i,j have been defined yet. I think they describe the detector sensitivity? Is this the same as QEij?

P7 Equation 10 – Consider using a different symbol for the gain, as G is already used for the STF.

P8 L1 – the straylight IS ASSOCIATED with general illumination conditions (e.g. solar angles) AND it will vary with time.

P8 L21 – Do you mean that the ACCURACY is limited by the difficulty in identifying plume-free pixels?

P9 L28 - . . . campaign IN which we participated. . .

P9 L30 - . . . two types of SITES:. . .

P9 L33 - . . . of which more than 93% IS GENERATED FROM coal.

P10 L4 - . . . close to ideal an BEST illustrate. . .

P11 L1 - . . . image areas UNAFFECTED by the plume.

P13 L4 - . . . camera cannot DISTINGUISH their respective SIGNATURES.

P13 L16 – SCDs are retrieved from at least two images TAKEN AT WAVELENGTHS

where absorption. . .

P13 L20 – A mathematical FRAMEWORK FOR DATA EVALUATION has been developed. . .

P13 L22 - .. offering longer INTEGRATION TIMES or more . . .

References – Please check the references for errors in displaying subscripts, particularly with the words SO2 and TeO2.

---

## Referee Comment (RC2) · C. Kern (Referee) · 23 Sep 2016

In reading more about the use of AOTF in atmospheric research, I came across the following paper which I think includes a lot of useful information and could be referenced in the final version of your manuscript (perhaps in the introduction?)

Cheng, A., Chan, M.H., 2004. Acousto-Optic Differential Optical Absorption Spectroscopy for Atmospheric Measurement of Nitrogen Dioxide in Hong Kong. Appl. Spectrosc. 58, 1462–1468.

Thank you for your consideration.
* * *

---

## Referee Comment (RC3) · J.-F. Smekens (Referee) · 16 Oct 2016

General comments

This manuscript describes the development of a new imaging instrument specifically designed to retrieve Slant Column Densities (SCD) of Nitrogen Dioxide (NO2) in gas plumes. The technique relies on the use of an Acousto-Optical Tunable Filter (AOTF) to create image pairs of a scene at two very close wavelengths showing a strong difference in the absorption cross-section of NO2 in the visible blue spectrum. The method is very innovative and offers a wide range of application for pollution monitoring, as well as for volcanological applications. I strongly recommend the publication of this manuscript and have only a few specific comments and recommendations that could improve the general discussion and its relevance to the volcanological community.

Specific comments

P2, L10 – It is certainly true there have been many developments in the SO2 camera technique in the last decade. However I would remain cautious in stating that the main technique for volcanological applications has shifted to SO2 cameras. Most observatories still rely on scanning spectrometers and the DOAS technique for the quantification of fluxes, and there have been great advances in the temporal resolution of scanning spectrometer arrays as well. To this day, very few SO2 cameras are used for continuous monitoring, in part due to the large amount of data they generate.

P8 - I would recommend a clarification as to the temporal resolution of the instrument, and that introduced by the mathematical methodology. Perhaps a more complete description of the averaging technique can be included in paragraph 3.3, stating the wavelengths used in the following example, exactly how long the exposure time were (0.5s?), how long it takes to acquire the 8 image pairs, and how many images were used to achieve an acceptable SNR. In the discussion, I also recommend adding a brief paragraph about the effect of exposure time on the SNR, and whether using fewer image pairs taken with a longer exposure time would help increase the temporal resolution. I realize the subject is mentioned several times throughout the manuscript, but sometimes with conflicting information.

P9, L25 – Here you state that the plume must remain optically thin for a successful application of the method. Indeed, heavy condensation of a plume near the vent has caused significant interference when using SO2 cameras, usually leading to the underestimation of the SCD. Although the AOTF method operates differently, a similar effect could be taking place here and influence your conclusions about the rate of conversion of NO to NO2. Could you comment on the optical thickness of the plumes in your example, and perhaps include a photograph of the scene?

P12 – Here you discuss the dynamic behaviour of the plume and the conversion of NO to NO2. It could be useful to include a time series of fluxes or concentrations taken at a given height above the stacks, in order to explore the dynamic nature of the plume. My understanding is that the temporal resolution of 3 minutes is due to the averaging of

multiple pairs of consecutive images. It occurs to me that a lower temporal resolution could be achieved by using a moving average technique.

P11 – On the use of this new instrument for the correction of SO2 camera data, your point is well taken. It would be quite valuable to employ this technique to correct SO2 camera measurements. However, the (near) simultaneity of acquisition of images in both wavelengths is in my opinion one of the biggest advantages of the SO2 camera, and one that allows us to truly investigate dynamic processes on very short timescales. Not only is the temporal resolution of the AOTF NO2 camera much lower, it provides an average of many images over 3 minutes, which will be difficult to reconcile with images from an SO2 camera produced at a rate of over 1 Hz. I would suggest adding a brief discussion on the practical application of such a correction OF SO2 camera data using the AOTF instrument.

The averaging of multiple image pairs is a strong limitation of the method presented here. Based on the images presented in figure 5, it seems they were acquired over a period of over 3 minutes. With a plume velocity of 5 m s-1, and based on the size of the field of view, it occurs to me that features image at the beginning of the acquisition will have moved outside of the FOV long before the end of the acquisition. If the plume is unsteady, and the emission rate varies during this 3 minute period (I strongly suspect that it does), this could have a significant impact on the interpretation of the SCD maps. Perhaps this is due to confusion in my understanding of the averaging process. But if it is not, could you clarify how this will affect your interpretation of the images, the calculation of the emission rates and the associated errors?

Technical corrections

I have not found any technical corrections beyond those already identified by Reviewer 1.

---

## Author Comment (AC1) · 16 Nov 2016

Dear Dr Kern,
Thank you for the positive recommendation and the in-depth review of our manuscript. Your thorough analysis greatly improved a number of weaknesses still present in the discussion version. Here below we address all your questions/comments, hoping that our answers will meet your expectancies.

- p.2,L3: U. Platt's reference for DOAS.
Agreed, the reference to Lohberger et al. (2004) has been replaced by Platt et al (1979).

- p.2,L9: Potential advantage of DOAS grating spectrometer over NO2 camera.
As written in the text (p.2, L6-9), there is no qestioning of the performance of the classical DOAS approach regarding detection limit or accuracy. Clearly, the NO2 camera

concept still needs further development and use before it approaches the same level of performance. The clear advantage of the NO2 camera over DOAS imaging systems is with the spatio-temporal resolution and the integrity of the data product (maps of NO2 SCDs). Even if the NO2 camera must degrade its instantaneous temporal resolution (exposure times of typically 0.5-1 second) by performing data averaging in order to mitigate plume transient features, the overall plume is imaged every time. The retrieved NO2 SCD map truly represents the mean SCD field. This is not the case with scanning spectrometers because only a small portion of the scene is probed at a time. Transient features in the plume not observed by the DOAS instrument at a given time can significantly affect the mean image. We think the current version of the manuscript is reflecting this position without trying to prove that the NO2 camera is a better instrument in general.

- p.2,L11: Disagreement with the statement on the advent of SO2 camera as main volcanic SO2 measurement technique.
Point taken, the sentence will be changed to: "In volcanology for instance, the so-called SO2 cameras are now increasingly complementing the measurements performed with classical dispersive techniques (grating spectrometers). Their concept ..."

- p.3,L9: Possibility of having all light rays travelling parallel through the AOTF.
We confirm that it is feasible to make all chief rays travel parallel through the AOTF. This is the role of the telecentric design: selecting only the chief rays and a very narrow cone of surrounding beams. It is achieved by placing a pupil at the focal plane of the first lens. This is crucial for preserving the spectral purity of the image. By ensuring that all rays hit the AOTF crystal surface with the same angle (+/- tolerance), the same wavelength will be selected across the field of view. On the other side, it is true that this design reduces the throughput of the instrument. But one cannot waive the physical principles of the AOTF such that the telecentricity must be obeyed within the tolerances of the acceptance angle of the AOTF.

- p.3,L21: Dependence of the acoustic power on the optical wavelength.

The efficiency of the acousto-optic (AO) interaction relies on the coupling between the light electric field and the elastic modulations created by the acoustic wave (which perturbate the dielectric susceptibility of the medium). The ease of the coupling depends on the physical properties of the crystal: refractive indices ($n_o$, $n_e$), elasto-optic coefficient ($p$), mass density ($\rho$), acoustic wave phase velocity ($v$). In acousto-optics, a figure of merit is often used to quickly assess which material is good for AO: $M_2 = n_e^3 n_o^3 p^2 / (\rho v^3)$. As an example, TeO2 is generally the preferred choice in VIS-SWIR applications as its $M_2$ is several orders of magnitude higher than the $M_2$ of quartz. As a consequence, a TeO2-based AOTF requires much less acoustic power than other materials for the same efficiency. The reason why it changes with wavelength is because of the dependence on the refractive indices.

- p.4,L5: Resolvable spots.
The number of resolvable spots is a relatively common concept in imaging optics which has to do with the modulation transfer function (MTF), i.e. the capability for the imaging system to resolve sharp contrasts (and not blur edges or lines). This is a purely spatial concept. As other optical parts, the AOTF is not capable of infinite spatial resolution. Besides the basic purity of the crystal, the divergence of the optical beam is also playing a role. Propagation angles are crucial in an AOTF: within the narrow cone of light surrounding a chief ray, the slight divergence of the beams causes a slight difference of diffraction angle, ending up with a decrease of the imaging quality. This effect is particularly true in the plane of the AO interaction, this is why the number of resolvable spots is different in both directions. The paper cited in the beginning of the paragraph (p.3,L25: Voloshinov et al. 2007) is a good reference for all these concepts.

- p.7,L7: Wavelength selection.
The reasons driving the selection of the two wavelengths are discussed in the first two paragraphs of section 3 on page 4: it is a matter of maximizing the differential optical depth while minimizing the spectral interval to avoid interference by aerosols or Rayleigh scattering.

This could be made clearer by changing the text surrounding eq.8 and 9. Starting at p.7, L6, the new text will be:

"If the spectral interval between $\lambda_w$ and $\lambda_s$ is small enough that the approximation $\tau_\star(\lambda_w) = \tau_\star(\lambda_s)$ holds, then the ratio of the transmittances $T(\lambda_w)/T(\lambda_s)$ is a measured quantity which only depends on the NO2 content of the plume. Introducing the relative instrument response at pixel $ij$: $\rho_{ij}(\lambda) = r_{ij}(\lambda)/r(\lambda)$, we find:

EQ.(8).

Finally, the NO2 SCD subtended by the area of the plume observed by pixel $ij$ follows by taking the logarithm of the ratio of transmittances:

EQ.(9).

Clearly, the best sensitivity is reached by maximizing the differential optical thickness when selecting $\lambda_w$ and $\lambda_s$."

- p.9,L14: Temperature effect on the AOTF passband.
You are perfectly correct: drifts in crystal temperature displace the central wavelength of the AOTF passband. At the time of the reported experiment, the driving electronics was not capable of adjusting the acoustic frequency to compensate for the drift in temperature. This led us to live with sub-optimal values of differential optical thicknesses. However, the temperature was monitored during the experiment which allowed for re-calibration of the wavelength scale during the post-measurement processing. Thanks to that, a maximum of 10% error is accounted for in the error budget: because the final wavelength uncertainty is about 0.1nm, which ends up with changes of not more than 10% in this region of the NO2 spectrum.

- p.11,L1: The relative error of 0.5%.
The relative error we are mentioning here is on the background signal $C_0$, not on the NO2 SCD. This value of 0.5% is in line with the discussion in the third paragraph of section 3.4 which deals with the various error sources. It was estimated from the sample statistics that served to compute the average value for $C_0$.
The sentence will be changed to make it clearer: "The relative error on $C_0$ is about

0.5% ...".

- p.12: Optical thickness of the plume, presence of aerosols.
During the measurement campaign, the content of the exhaust plume often changed, sometimes within a few seconds. Actually, most of the time, the smokes were white and opaque, possibly caused by some smoke washing process. In this paper, we are only showing results applicable to optically thin smokes. The sample results illustrating the paper (fig.5) have been taken in this situation. The smokes were slightly brownish while clouds could also be observed passing behind. By comparing the background signal with the plume signal, it appears that the plume optical thickness at the measurement wavelengths was around 0.06-0.08 above the 2nd stack, and 0.04-0.06 above the 3rd stack.
This point will be added in the text with a new sentence at the end of p.10,L4: "In particular, the smokes were optically thin, with the blue sky clearly visible in the background. This ensures that absorption is the dominant process over scattering for the extinction of light rays crossing the plumes (Beer-Lambert regime). The optical thickness of the smokes was always smaller than 0.1 at our measurement wavelengths."

Technical corrections:

Unless for the particular points discussed below, all the suggested technical corrections have been accepted.

- p.2,L31: "atmospheric constituents" is changed to "atmospheric species concentration profiles"

- p.3,L3: "breadboard" is a shorthand name for a lab optical setup used to prove a

concept. But we understand that readers may not be familiar with this convention so we change all occurences of "breadboard" by "prototype".

- p.5,Fig.3: "Measured" replaced by "NO2 camera".

- p.7,L13: rho is defined in p.7,L8. It represents the instrument response at pixel ij embracing the effects of the optics, the AOTF and the detector, relative to an average value.

---

## Author Comment (AC2) · 16 Nov 2016

Dear Dr Kern,
Thank you for drawing our attention to the paper by Cheng and Chan (2004). We were aware of a later paper by the same authors:
Cheng and Chan (2005). Acousto-optic measurements of tropospheric nitrogen dioxide column density by solar spectroscopy. Applied Optics, 44(26), 5536–5543.
The same concept is exploited in both papers (2004 and 2005): use of an AOTF to measure a broadband solar spectrum through the atmosphere, then apply the standard DOAS retrieval method. According to us, this is not a very clever use of an AOTF in remote sensing: a considerable amount of time is spent scanning the spectrum wavelength by wavelength to obtain what can be achieved within a second with a grating-based solution. Moreover, the imaging capability of the AOTF is not exploited whereas it is the key advantage compared to scanning systems.

[Figure]

We would prefer not to mention these papers because we would have to spend several sentences explaining why it is not such a good example of using AOTF technology in atmospheric remote sensing.

---

## Author Comment (AC3) · 16 Nov 2016

**Dear Dr Smekens,**

Thank you for the positive recommendation and the time taken to perform the in-depth review of our manuscript. Your comments and questions have contributed to improving the quality of our paper. Please find below our answers to your questions and comments.

- p.2,L10: Relative importance of SO2 cameras in volcanic plumes remote sensing. Point taken, the sentence starting on p.2,L10 will be changed to: "In volcanology for instance, the so-called SO2 cameras are now complementing the measurements performed with classical dispersive techniques (grating spectrometers). Their concept ..."

- p.8: Data averaging, exposure time and SNR.

Section 3.3 on p.8 is devoted to detail how, mathematically-speaking, multiple images taken at the same wavelength can be combined; and how a retrieval can also be attempted based on multiple image pairs. This chapter is really about mathematical formalism only.

The timing of the acquisitions reported in this manuscript are given in the first paragraph of section 4.1. It is stated that the exposure times for all wavelengths was 0.5 second, and a dwell time of 1.3 second must be accounted for between each image. In total, 13 seconds were needed to complete the series of 8 spectral images. Eventually, it turned out that the four shortest wavelengths were delivering too noisy images (because of the weaker natural radiance and the instrument sensitivity droping towards the blue), whereas the four longest ones gave decent measurements. As explained in the second paragraph of section 4.1, the essential reason why we had to average the images is because of the plume dynamics. From successive trials, we found that starting from 10 loops averaged (i.e. 10 times the same 4 wavelengths reduced to 4 images), plume mismatches between successive snapshots had essentially vanished. The sentence on p.10,L13 will be extended: "The dwell time between the closing of the shutter and its re-opening was 1.3 s, yielding a total acquisition sequence duration of 13.1 s for the 8 spectral images."

- p.9,L25: Optical thickness of the plume.

During the measurement campaign, the content of the exhaust plume often changed, sometimes within a few seconds. Actually, most of the time, the smokes were white and opaque, possibly caused by some smoke washing process. In this paper, we are only showing results applicable to optically thin smokes. The results samples illustrating the paper (fig.5) have been taken in this situation. The smokes were slightly brownish while clouds could also be observed passing behind. By comparing the background signal with the plume signal, it appears that the plume optical thickness at the measurement wavelengths was around 0.06-0.08 above the 2nd stack, and 0.04-0.06 above the 3rd stack.

This point will be added in the text with a new sentence at the end of p.10,L4: "In partic-

AMTD
ular, the smokes were optically thin, with the blue sky clearly visible in the background. This ensures that absorption is the dominant process over scattering for the extinction of light rays crossing the plumes (Beer-Lambert regime). The optical thickness of the smokes was always smaller than 0.1 at our measurement wavelengths."

**- p.12: Time series of NO2 fluxes.**

Indeed, we could plot the evolution of the NO2 flux at some reference altitude above the stacks. To do so, we have planned to use the moving average technique as you suggest. These kinds of results will be exploited in another paper belonging to the AROMAT campaigns special issue (in AMT as well). In that paper, we hope to go a little bit deeper into the plume chemistry. Our feeling was that the NO2 SCD maps, plus a preview of what can be extracted from them (the NO2 flux profile), is sufficient in a paper whose scope is to present a new instrument...

- p.11: Potential capability of correcting SO2 camera data.

We suppose that you are actually referring to the first paragraph on p.13.

You are correct to point out that this first NO2 camera suffers from a poorer temporal resolution compared to that achieved by SO2 cameras. If we hadn't wasted time with taking images at the 4 shorter wavelengths, we would have been capable of providing NO2 SCD maps every 1.5 minutes or so. We are currently working on vastly improving this aspect, and we have good hopes to end up with a system capable of delivering SCD maps every 10 seconds in the same illumination conditions. However, we will probably never reach 1Hz because the throughput of the instrument is smaller than for a SO2 camera.

We will add the following sentence on p.13,L10: "On the temporal resolution side though, the NO2 camera is, at the moment, not capable of following the pace of SO2 cameras (1 Hz typical), such that the correction maps would have to be applied to temporally-averaged SO2 data."

- Limitations of data averaging.

We are not sure we understand the point here... As you write it, plume transient fea-
tures make the NO2 field unsteady. This is why we were forced to average our acquisitions over time in order to work on a "mean" plume, rather than on the instantaneous plume which turned out to be impractical with our sequential acquisitions. To come back to the numbers presented in section 4.1: the 12 sequences of 4 wavelengths constitute 48 samples of the scene over 3 minutes. To put it differently, the plumes were observed every 3.75 seconds at each wavelength during 3 minutes. We first combined the samples wavelength-wise. This gave a "mean" scene observed at 4 wavelengths. Two doublets were created from these 4 "mean" images and finally, the NO2 SCD map was retrieved. If the "mean" image wouldn't have been consistent from one wavelength to another, then artefacts like negative SCDs would have appeared in the final result. This is precisely based on the absence of false negative SCDs that we found that at least 10 sequences had to be averaged.

Maybe the point of misunderstanding is that we are not calculating NO2 SCDs based on single pairs of images, but on the "mean" images... By comparison, we suspect that much more doubts can be raised with the SCD maps obtained with scanning DOAS instruments.